# SELF-REFLECTIVE VARIATIONAL AUTOENCODER

## ABSTRACT

The Variational Autoencoder (VAE) is a powerful framework for learning probabilistic latent variable generative models. However, typical assumptions on the approximate posterior distributions can substantially restrict its capacity for inference and generative modeling. Variational inference based on neural autoregressive models respects the conditional dependencies of the exact posterior, but this flexibility comes at a cost: the resulting models are expensive to train in high-dimensional regimes and can be slow to produce samples. In this work, we introduce an orthogonal solution, which we call *self-reflective inference*. By redesigning the hierarchical structure of existing VAE architectures, self-reflection ensures that the stochastic flow preserves the factorization of the exact posterior, sequentially updating the latent codes in a manner consistent with the generative model. We empirically demonstrate the advantages of matching the variational posterior to the exact posterior—on binarized MNIST self-reflective inference achieves state-of-the-art performance without resorting to complex, computationally expensive components such as autoregressive layers. Moreover, we design a variational normalizing flow that employs the proposed architecture, yielding predictive benefits compared to its purely generative counterpart. Our proposed modification is quite general and it complements the existing literature; self-reflective inference can naturally leverage advances in distribution estimation and generative modeling to improve the capacity of each layer in the hierarchy.

## 1 INTRODUCTION

The advent of deep learning has led to great strides in both supervised and unsupervised learning. One of the most popular recent frameworks for the latter is the Variational Autoencoder (VAE), in which a probabilistic encoder and generator are jointly trained via backpropagation to simultaneously perform sampling and variational inference. Since the introduction of the VAE (Kingma & Welling, 2014), or more generally, the development of techniques for low-variance stochastic backpropagation of Deep Latent Gaussian Models (DLGMs) (Rezende et al., 2014), research has rapidly progressed towards improving their generative modeling capacity and/or the quality of their variational approximation. However, as deeper and more complex architectures are introduced, care must be taken to ensure the correctness of various modeling assumptions, whether explicit or implicit. In particular, when working with hierarchical models it is easy to unintentionally introduce mismatches in the generative and inference models, to the detriment of both. In this work, we demonstrate the existence of such a modeling pitfall common to much of the recent literature on DLGMs. We discuss why this problem emerges, and we introduce a simple—yet crucial—modification to the existing architectures to address the issue.

Vanilla VAE architectures make strong assumptions about the posterior distribution—specifically, it is standard to assume that the posterior is approximately factorial. More recent research has investigated the effect of such assumptions which govern the variational posterior (Wenzel et al., 2020) or prior (Wilson & Izmailov, 2020) in the context of uncertainty estimation in Bayesian neural networks. In many scenarios, these restrictions have been found to be problematic. A large body of recent work attempts to improve performance by building a more complex encoder and/or decoder with convolutional layers and more modern architectures (such as ResNets (He et al., 2016)) (Salimans et al., 2015; Gulrajani et al., 2017) or by employing more complex posterior distributions constructed with autoregressive layers (Kingma et al., 2016; Chen et al., 2017). Other work (Tomczak & Welling, 2018; Klushyn et al., 2019a) focuses on refining the prior distribution of the latent

codes. Taking a different approach, hierarchical VAEs (Rezende et al., 2014; Gulrajani et al., 2017; Sønderby et al., 2016; Maaløe et al., 2019; Klushyn et al., 2019b) leverage increasingly deep and interdependent layers of latent variables, similar to how subsequent layers in a discriminative network are believed to learn more and more abstract representations. These architectures exhibit superior generative and reconstructive capabilities since they allow for modeling of much richer latent spaces. While the benefits of incorporating hierarchical latent variables is clear, all existing architectures suffer from a modeling mismatch which results in sub-optimal performance: *the variational posterior does not respect the factorization of the exact posterior distribution of the generative model.*

In earlier works on hierarchical VAEs (Rezende et al., 2014), inference proceeds bottom-up, counter to the top-down generative process. To better match the order of dependence of latent variables to that of the generative model, later works (Sønderby et al., 2016; Bachman, 2016) split inference into two stages: first a deterministic bottom-up pass which does necessary precomputation for evidence encoding, followed by a stochastic top-down pass which incorporates the hierarchical latents to form a closer variational approximation to the exact posterior. Crucially, while these newer architectures ensure that the order of the latent variables mirrors that of the generative model, the overall variational posterior does not match because of the strong restrictions on the variational distributions of each layer.

**Contributions.**   In this work, we propose to restructure common hierarchical VAE architectures with a series of bijective layers which enable communication between the inference and generative networks, refining the latent representations. Concretely, our contributions are as follows:

- We motivate and introduce a straightforward *rearrangement of the stochastic flow of the model* which addresses the aforementioned modeling mismatch. This modification substantially compensates for the observed performance gap between models with only simple layers and those with complex autoregressive networks (Kingma et al., 2016; Chen et al., 2017).

- We formally prove that this refinement results in **a hierarchical VAE whose variational posterior respects the precise factorization of the exact posterior.** To the best of our knowledge, this is the first deep architecture to do so without resorting to computationally expensive autoregressive components or making strong assumptions (e.g., diagonal Gaussian) on the distributions of each layer (Sønderby et al., 2016)—assumptions that lead to degraded performance.

- We experimentally demonstrate the benefits of the improved representation capacity of this model, which stems from the corrected factorial form of the posterior. We achieve state-of-the-art perfomance on MNIST among models without autoregressive layers, and our model performs on par with recent, fully autoregressive models such as Kingma et al. (2016). Due to the simplicity of our architecture, we achieve these results for a fraction of the computational cost in both training and inference.

- We design a *hierarchical variational normalizing flow* that deploys the suggested architecture in order to recursively update the base distribution and the conditional bijective transformations. This architecture significantly improves upon the predictive performance and data complexity of a Masked Autoregressive Flow (MAF) (Papamakarios et al., 2017) on CIFAR-10.

Finally, it should be noted that our contribution is quite general and can naturally leverage recent advances in variational inference and deep autoencoders (Chen et al., 2017; Kingma et al., 2016; Tomczak & Welling, 2018; Burda et al., 2016; Dai & Wipf, 2019; van den Oord et al., 2016a; Rezende & Viola, 2018) as well as architectural improvements to density estimation (Gulrajani et al., 2017; Dinh et al., 2017; Kingma & Dhariwal, 2018; Durkan et al., 2019; van den Oord et al., 2016b; Gregor et al., 2015). We suspect that combining our model with other state-of-the-art methods could further improve the attained performance, which we leave to future work.

## 2   VARIATIONAL AUTONENCODERS

A Variational Autoencoder (VAE) (Kingma & Welling, 2014; 2019) is a generative model which is capable of generating samples $x \in \mathbb{R}^D$ from a distribution of interest $p(x)$ by utilizing latent variables $z$ coming from a prior distribution $p(z)$. To perform inference, the marginal likelihood

should be computed which involves integrating out the latent variables:

$$p(\boldsymbol{x}) = \int p(\boldsymbol{x}, \boldsymbol{z}) \, d\boldsymbol{z}. \tag{1}$$

In general, this integration will be intractable and a lower bound on the marginal likelihood is maximized instead. This is done by introducing an approximate posterior distribution $q(\boldsymbol{z} \mid \boldsymbol{x})$ and applying Jensen's inequality:

$$\log p(\boldsymbol{x}) = \log \int p(\boldsymbol{x}, \boldsymbol{z}) \, d\boldsymbol{z} = \log \int \frac{q(\boldsymbol{z} \mid \boldsymbol{x})}{q(\boldsymbol{z} \mid \boldsymbol{x})} p(\boldsymbol{x}, \boldsymbol{z}) \, d\boldsymbol{z} \geq \int q(\boldsymbol{z} \mid \boldsymbol{x}) \log \left[ \frac{p(\boldsymbol{x} \mid \boldsymbol{z}) p(\boldsymbol{z})}{q(\boldsymbol{z} \mid \boldsymbol{x})} \right] d\boldsymbol{z}$$

$$\implies \log p(\boldsymbol{x}) \geq \mathbb{E}_{q(\boldsymbol{z} \mid \boldsymbol{x})} [\log p(\boldsymbol{x} \mid \boldsymbol{z})] - D_{KL}(q(\boldsymbol{z} \mid \boldsymbol{x}) \parallel p(\boldsymbol{z})) \triangleq \mathcal{L}(\boldsymbol{x}; \boldsymbol{\theta}, \boldsymbol{\phi}), \tag{2}$$

where $\boldsymbol{\theta}$, $\boldsymbol{\phi}$ parameterize $p(\boldsymbol{x}, \boldsymbol{z}; \boldsymbol{\theta})$ and $q(\boldsymbol{z} \mid \boldsymbol{x}; \boldsymbol{\phi})$ respectively. For ease of notation, we may omit $\boldsymbol{\theta}, \boldsymbol{\phi}$ in the derivations. This objective is called the Evidence Lower BOund (ELBO) and can be optimized efficiently for continuous $\boldsymbol{z}$ via stochastic gradient descent (Kingma & Welling, 2014; Rezende et al., 2014).

## 3 SELF-REFLECTIVE VARIATIONAL INFERENCE

With this background, we are now ready to introduce our main contribution: the first deep probabilistic model which ensures that the variational posterior matches the factorization of the exact posterior induced by its generative model. We refer to this architecture as the Self-Reflective Variational Autoencoder (SeRe-VAE). We expound upon its components in the following subsections.

### 3.1 GENERATIVE MODEL

Figure 1 displays the overall stochastic flow of the generative network. A detailed illustration of our model is provided in Figure S3.

Our generative model consists of a hierarchy of $L$ stochastic layers, as in Rezende et al. (2014). However, in this work, the data $\boldsymbol{x} = (\boldsymbol{x}_1, \boldsymbol{x}_2, \ldots, \boldsymbol{x}_L) \in \mathbb{R}^D$ is partitioned into $L$ blocks, with each layer generating only $\boldsymbol{x}_l \in \mathbb{R}^{D_l}$, with $\sum_l D_l = D$. At each layer $l$, $N_l$-dimensional latent variables $\boldsymbol{\epsilon}_l \in \mathbb{R}^{N_l}$ are first sampled from a simple prior distribution (*prior layer*) and subsequently transformed to latent variables $\boldsymbol{z}_l \in \mathbb{R}^{N_l}$ by a bijective function $f_l : \mathbb{R}^{N_l} \to \mathbb{R}^{N_l}$.

To distinguish between the two sets of latent variables in our model, throughout this paper we refer to $\boldsymbol{\epsilon}_l$ as the *base latent variables* and $\boldsymbol{z}_l$ as the *latent codes*. For example, for an affine transformation $f_l$ the latent codes are given by $\boldsymbol{z}_l = f_l(\boldsymbol{\epsilon}_l) = \boldsymbol{c}_l + (diag(\boldsymbol{d}_l) + \boldsymbol{u}_l \boldsymbol{u}_l^T) \times \boldsymbol{\epsilon}_l$, with $\boldsymbol{c}_l, \boldsymbol{u}_l, \boldsymbol{d}_l \in \mathbb{R}^{N_l}$ and $\boldsymbol{d}_l \geq 0$ to ensure bijectivity. The latent codes $\boldsymbol{z}_l$ are subsequently passed to the stochastic layer responsible for generating the observed data $\boldsymbol{x}_l$ (*data layer*).

Moreover, the layers in the hierarchy are connected in three ways: i) the prior layer $l$ can access the latent codes $\boldsymbol{z}_{l-1}$ defining a conditional distribution $p(\boldsymbol{\epsilon}_l \mid \boldsymbol{z}_{l-1})$ ii) $\boldsymbol{z}_{l-1}$ is fed to the next bijection $f_l$ defining a conditional transformation $\boldsymbol{z}_l = f_l(\boldsymbol{\epsilon}_l \mid \boldsymbol{z}_{l-1})$ iii) the data layer $l$ receives the data block $\boldsymbol{x}_{l-1}$ generated by the previous data layer defining a conditional distribution $p(\boldsymbol{x}_l \mid \boldsymbol{z}_{l-1}, \boldsymbol{x}_{l-1})$. Intuitively, this choice is justified because the latent codes $\boldsymbol{z}_l$ of layer $l$, conditioned on $\boldsymbol{z}_{l-1}$, will be successively refined based on how well $\boldsymbol{z}_{l-1}$ reconstructed $\boldsymbol{x}_{l-1}$, yielding progressively more meaningful latent representations. In the following subsections, we describe these steps in detail. The joint distribution of the base latent variables $\boldsymbol{\epsilon} = (\boldsymbol{\epsilon}_1, \boldsymbol{\epsilon}_2, \ldots, \boldsymbol{\epsilon}_L)$ and the observed data $\boldsymbol{x}$ of the generative model is:

$$p(\boldsymbol{x}, \boldsymbol{\epsilon}) = p(\boldsymbol{\epsilon}_1) \times p(\boldsymbol{x}_1 | \boldsymbol{z}_1) \times \prod_{l=2}^{L} p(\boldsymbol{\epsilon}_l \mid \boldsymbol{z}_{l-1}) \times p(\boldsymbol{x}_l \mid \boldsymbol{z}_{l-1}, \boldsymbol{x}_{l-1}). \tag{3}$$

### 3.2 INFERENCE MODEL

The inference network is identical to the generative network shown in Figure 1, except that the prior layers are replaced by *posterior layers*, that are additionally conditioned on the observed data $\boldsymbol{x}$, for

Figure 1: *D-separation between stochastic layers.* By the Bayes ball rule, all paths from $\boldsymbol{\epsilon}_1$ to $\boldsymbol{\epsilon}_3$ pass either through $\boldsymbol{x}_1$ or $\boldsymbol{z}_2$, which $D$-separate them. Therefore, $\boldsymbol{\epsilon}_1 \perp\!\!\!\perp \boldsymbol{\epsilon}_3 | \boldsymbol{z}_2, \boldsymbol{x}$.

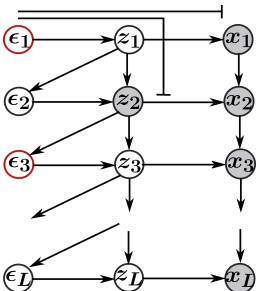

the generation of the base latent variables $\boldsymbol{\epsilon}_l$. Specifically, the variational encoder of the SeRe-VAE is defined as follows:

$$q(\boldsymbol{\epsilon}_1, \boldsymbol{\epsilon}_2, \ldots, \boldsymbol{\epsilon}_L \mid \boldsymbol{x}) = q(\boldsymbol{\epsilon}_1 \mid \boldsymbol{x}) \times \prod_{l=2}^{L} q(\boldsymbol{\epsilon}_l \mid \boldsymbol{z}_{l-1}, \boldsymbol{x}). \tag{4}$$

The formal justification of this factorization is deferred to section 3.3. Compared to other hierarchical architectures, in the proposed model the inference layers are conditioned on the output of the preceding bijective layer — these components are shared between the generative and the inference network (see also Figure S2). This choice allows for complex transformations of the latent variables and is theoretically motivated by the following proposition.

**Proposition 1** *Let $p(\boldsymbol{\epsilon})$ and $q(\boldsymbol{\epsilon})$ be two $N$-dimensional probability densities. Let $f : \mathbb{R}^N \to \mathbb{R}^N$ be an invertible, smooth transformation of the random variable $\boldsymbol{\epsilon}$ such that $\boldsymbol{z} = f(\boldsymbol{\epsilon})$, yielding distributions $p'(\boldsymbol{z})$ and $q'(\boldsymbol{z})$ of $\boldsymbol{z}$ respectively. Then, $D_{KL}(q'(\boldsymbol{z}) \parallel p'(\boldsymbol{z})) = D_{KL}(q(\boldsymbol{\epsilon}) \parallel p(\boldsymbol{\epsilon}))$.*

**Proof:** From the definition of the Kullback–Leibler divergence and the change of variables formula (Rudin, 2006; Bogachev, 2007):

$$\mathbb{E}_{q'(\boldsymbol{z})}\left[\log \frac{q'(\boldsymbol{z})}{p'(\boldsymbol{z})}\right] = \mathbb{E}_{q(\boldsymbol{\epsilon})}\left[\log \frac{q'(f(\boldsymbol{\epsilon}))}{p'(f(\boldsymbol{\epsilon}))}\right] = \mathbb{E}_{q(\boldsymbol{\epsilon})}\left[\log \frac{q(\boldsymbol{\epsilon}) \times \mid \det J_f(\boldsymbol{\epsilon}) \mid^{-1}}{p(\boldsymbol{\epsilon}) \times \mid \det J_f(\boldsymbol{\epsilon}) \mid^{-1}}\right] = \mathbb{E}_{q(\boldsymbol{\epsilon})}\left[\log \frac{q(\boldsymbol{\epsilon})}{p(\boldsymbol{\epsilon})}\right], \tag{5}$$

where $J_f(\boldsymbol{\epsilon})$ is the Jacobian matrix of $f$ evaluated at $\boldsymbol{\epsilon}$. $\square$

Proposition 1 implies that the inclusion of the bijectors $f_l$ can help increase the conditional likelihood $p(\boldsymbol{x} \mid \boldsymbol{z})$ in equation 2 without increasing the KL term. Moreover—though not pursed in this work—it motivates the construction of normalizing flows for variational inference with non-linear time determinant of the Jacobian matrix, since the analytical form of the transformed distribution is no longer needed for the computation of the KL-divergence. In this work, we assume Gaussian diagonal base distributions. In order to account for the two conditioning streams, the evidence $\boldsymbol{x}$ and the latent factors $\boldsymbol{z}_{l-1}$, we employ a residual parametrization as described in section 3.4.2.

### 3.3 EXACT BAYES PROPAGATION

In this section, we provide the formal justification for the choice of equation 4: we prove that backpropagation of our model preserves the factorization of the true posterior, without resorting to complex graph inversion as in Webb et al. (2018). We use the following straightforward lemma:

**Lemma 1** *Let $f : \mathbb{R}^N \to \mathbb{R}^N$ be an invertible transformation such that both $f$ and $f^{-1}$ are differentiable everywhere. Then for any $\boldsymbol{z} \in \mathbb{R}^N$, $p(\boldsymbol{\epsilon}|\boldsymbol{z}) = p(\boldsymbol{\epsilon}|f(\boldsymbol{z}))$.*

**Proof:** By Bayes's Theorem and the change of variables formula (Rudin, 2006; Bogachev, 2007),

$$p(\boldsymbol{\epsilon}|f(\boldsymbol{z})) = \frac{p(f(\boldsymbol{z})|\boldsymbol{\epsilon}) \times p(\boldsymbol{\epsilon})}{p(f(\boldsymbol{z}))} = \frac{p(\boldsymbol{z}|\boldsymbol{\epsilon}) \times \mid \det J_f(\boldsymbol{z}) \mid^{-1} \times p(\boldsymbol{\epsilon})}{p(\boldsymbol{z}) \times \mid \det J_f(\boldsymbol{z}) \mid^{-1}} = p(\boldsymbol{\epsilon}|\boldsymbol{z}),$$

where $J_f(\boldsymbol{z})$ is the Jacobian matrix of $f$ evaluated at $\boldsymbol{z}$, which has non-zero determinant by assumption. $\square$

We now present our main theoretical result, which says that the factorization of our model's variational posterior exactly matches that of the generative distribution.

**Proposition 2** *The factorization of the variational posterior defined in equation 4 respects the factorization of the exact posterior distribution induced by the generative model in equation 3.*

**Proof:** Let $p(\boldsymbol{\epsilon}_1, \boldsymbol{\epsilon}_2, \ldots, \boldsymbol{\epsilon}_L \mid \boldsymbol{x})$ be the posterior distribution induced by the generative model defined in equation 3, as illustrated in Figure 1. Then, according to the probability product rule the posterior distribution can be expressed as:

$$p(\boldsymbol{\epsilon}_1, \boldsymbol{\epsilon}_2, \ldots, \boldsymbol{\epsilon}_L \mid \boldsymbol{x}) = p(\boldsymbol{\epsilon}_1 \mid \boldsymbol{x}) \times \prod_{l=2}^{L} p(\boldsymbol{\epsilon}_l \mid \boldsymbol{\epsilon}_{<l}, \boldsymbol{x}), \tag{6}$$

where $\boldsymbol{\epsilon}_{<l} \triangleq \{\boldsymbol{\epsilon}_1, \boldsymbol{\epsilon}_2, \ldots, \boldsymbol{\epsilon}_{l-1}\}$. We will apply the *Bayes ball* rule (Jordan, 2003) to simplify equation 6. Consider an arbitrary layer $l$ of the hierarchy. Because $f_{l-1}$ is a bijector, by Lemma 1 we have

$$p(\boldsymbol{\epsilon}_l \mid \boldsymbol{\epsilon}_{<l}, \boldsymbol{x}) = p(\boldsymbol{\epsilon}_l \mid \boldsymbol{\epsilon}_{l-1}, \boldsymbol{\epsilon}_{<l-1}, \boldsymbol{x}) = p(\boldsymbol{\epsilon}_l \mid \boldsymbol{z}_{l-1}, \boldsymbol{\epsilon}_{<l-1}, \boldsymbol{x}).$$

Now, note that $\boldsymbol{\epsilon}_l$ is *D-separated* from $\boldsymbol{\epsilon}_{l-1}, \ldots, \boldsymbol{\epsilon}_1$ since all paths from $\boldsymbol{\epsilon}_l$ to $\boldsymbol{\epsilon}_{<l}$ pass through the observed nodes $\boldsymbol{z}_{l-1}$ or $\boldsymbol{x}_1, \boldsymbol{x}_2, \ldots, \boldsymbol{x}_{l-1}$ (see Figure 1 for an example). Therefore, we have

$$p(\boldsymbol{\epsilon}_l \mid \boldsymbol{z}_{l-1}, \boldsymbol{\epsilon}_{<l-1}, \boldsymbol{x}) = p(\boldsymbol{\epsilon}_l \mid \boldsymbol{z}_{l-1}, \boldsymbol{x}). \tag{7}$$

Since this applies to every layer, it follows that the exact posterior equation 6 can also be expressed as

$$p(\boldsymbol{\epsilon}_1, \boldsymbol{\epsilon}_2, \ldots, \boldsymbol{\epsilon}_L \mid \boldsymbol{x}) = p(\boldsymbol{\epsilon}_1 \mid \boldsymbol{x}) \times \prod_{l=2}^{L} p(\boldsymbol{\epsilon}_l \mid \boldsymbol{z}_{l-1}, \boldsymbol{x}), \tag{8}$$

exactly matching the factorization of the approximate posterior in equation 4. $\qquad\square$

### 3.4 IMPLEMENTATION DETAILS

#### 3.4.1 AMORTIZED LAYERS

We use an amortized parametrization to construct the conditional probability densities involved in the derivations above. In particular, for a probability density $p(\boldsymbol{\epsilon} \mid \boldsymbol{z}; \boldsymbol{\theta})$ we take the parametrization $\boldsymbol{\theta}$ as a function of $\boldsymbol{z}$: $\boldsymbol{\theta} \equiv \boldsymbol{\theta}(\boldsymbol{z})$. For example, a conditional Gaussian distribution is defined as $p(\boldsymbol{\epsilon} \mid \boldsymbol{z}) = \mathcal{N}(\boldsymbol{\mu}(\boldsymbol{z}), \boldsymbol{\sigma}(\boldsymbol{z}))$ with $\boldsymbol{\theta}(\boldsymbol{z}) = (\boldsymbol{\mu}(\boldsymbol{z}), \boldsymbol{\sigma}(\boldsymbol{z}))$. The computational graph of an amortized Gaussian layer is shown in Figure S4. Similarly, for a conditional bijector $f(\boldsymbol{\epsilon} \mid \boldsymbol{z}; \boldsymbol{\beta})$ we take $\boldsymbol{\beta}$ as a function of $\boldsymbol{z}$: $\boldsymbol{\beta} \equiv \boldsymbol{\beta}(\boldsymbol{z})$. For example, for the affine bijector defined in section 3.1, we consider $\boldsymbol{\beta}(\boldsymbol{z}) = (\boldsymbol{c}(\boldsymbol{z}), \boldsymbol{d}(\boldsymbol{z}), \boldsymbol{u}(\boldsymbol{z}))$.

#### 3.4.2 RESIDUAL DISTRIBUTIONAL LAYERS

All but the first data layer $p(\boldsymbol{x}_l \mid \boldsymbol{z}_{l-1}, \boldsymbol{x}_{l-1})$ and posterior layer $q(\boldsymbol{\epsilon}_l \mid \boldsymbol{z}_{l-1}, \boldsymbol{x})$ receive two streams of conditioning factors—one latent and one observed. We ensure that each factor incrementally refines the distribution by adopting a residual parametrization. Here we describe the residual Gaussian distribution when conditioned on the two factors $\boldsymbol{z}, \boldsymbol{x}$. Its probability density is given by

$$q(\boldsymbol{\epsilon} \mid \boldsymbol{z}, \boldsymbol{x}) = \mathcal{N}(\boldsymbol{\mu}(\boldsymbol{z})\delta\boldsymbol{\sigma}(\boldsymbol{x}) + \delta\boldsymbol{\mu}(\boldsymbol{x}), \boldsymbol{\sigma}(\boldsymbol{z})\delta\boldsymbol{\sigma}(\boldsymbol{x})), \tag{9}$$

which can be interpreted as follows. The first distribution $\mathcal{N}(\boldsymbol{\mu}(\boldsymbol{z}), \boldsymbol{\sigma}(\boldsymbol{z}))$ is corrected by the residuals $\delta\boldsymbol{\sigma}(\boldsymbol{x})$, $\delta\boldsymbol{\mu}(\boldsymbol{x})$; here we see the dependence on the conditioning factor $\boldsymbol{x}$. If $\boldsymbol{x}$ does not provide additional information on $\boldsymbol{\epsilon}$ (formally, $p(\boldsymbol{\epsilon} \mid \boldsymbol{z}, \boldsymbol{x}) = p(\boldsymbol{\epsilon} \mid \boldsymbol{z})$), the two corrections collapse to 1 and 0 respectively—that is, inducing no change. The reader may refer to Figure S7 where we qualitatively illustrate the effect of the residual distributional layer that improves the conditional likelihood provided by the first one. To reduce the number of parameters, we consider networks for $\boldsymbol{\mu}(\boldsymbol{z})$, $\boldsymbol{\sigma}(\boldsymbol{z})$ that are shared between the prior and the posterior, yielding a prior of the form $p(\boldsymbol{\epsilon} \mid \boldsymbol{z}) = \mathcal{N}(\boldsymbol{\mu}(\boldsymbol{z}), \boldsymbol{\sigma}(\boldsymbol{z}))$. Finally, we found experimentally that enforcing $\delta\boldsymbol{\sigma}(\boldsymbol{x}) \leq 1$ helps optimization by ensuring that $\boldsymbol{x}$ can only reduce the variance of the prior.

### 3.5 GENERAL REMARKS

Following the above analysis, we make some observations about the hierarchy of shared bijective layers in the model:

- In contrast to Rezende et al. (2014) (see Figure S1), in our model i) the prior layers are not independent, but rather are conditioned on the previous layers in the hierarchy; and ii) the transformational layers are restricted to be bijective.

- The proposed model also differs from other hierarchical architectures (Gulrajani et al., 2017; Sønderby et al., 2016; Maaløe et al., 2019); in these models the layers of the prior are conditioned upon the previous prior layers and not upon bijective layers that are shared between the generative and inference model.

- One additional key difference between our model and all previous work is the coupling between the data layers. Therefore, the decoder can be perceived layer-wise instead of pixel-wise autoregressive rendering the sampling much more efficient ($\mathcal{O}(L)$ instead of $\mathcal{O}(D)$). In section 4, we provide empirical results demonstrating the benefits of these modeling choices.

- By reducing the set of conditioning variables from $\epsilon_{<l}$ to $z_{l-1}$ in a theoretically justified manner, the hierarchical bijective layers offer a convenient way to precisely and efficiently factorize the variational distribution, alleviating the bottleneck present in high-dimensional autoregressive approaches.

- The model, albeit hierarchical, is less prone to posterior collapse, since each layer is responsible for the generation of a different portion of the data. Experimental support for this observation is provided in Figure S8, where we plot the KL divergence for each layer of the architecture investigated in section 4.1.2.

## 4 EXPERIMENTAL STUDIES

### 4.1 DYNAMICALLY BINARIZED MNIST

We empirically evaluate the SeRe-VAE on dynamically binarized MNIST. As in Burda et al. (2016); Sønderby et al. (2016); Kingma et al. (2016), the binary-valued observations are sampled after each epoch with the Bernoulli expectations being set equal to the real, normalized pixel values in the dataset which prevents overfitting.

### 4.1.1 PERFORMANCE OF THE MLP SERE-VAE

To demonstrate that our model's improved performance is due to the restructuring of the stochastic flow and not sophisticated layers, we use simple multilayer-perceptron (MLP) components; we similarly forgo importance weighting (Burda et al., 2016). We adopt a 10-layer architecture, with $N_l = 10$ latent variables per layer, for a total of 100 latent features being passed to the decoder after being transformed by an affine bijector as described in section 3.1. We partition the image into $L = 10$ equally sized blocks (except for the last one) from left to right in a raster fashion. Finally, we use independent deterministic encoders for the data preprocessing. The full details of our implementation are delegated to the supplementary material. We again emphasize the overall simplicity of our architecture, choosing instead to focus on the benefits of the corrected posterior factorization. As shown in Table 1, our model (SeRe-VAE) outperforms existing models of the same complexity

such as the DLGM and Ladder VAE (LVAE), those of higher complexity such as Inverse Autoregressive Flow (IAF), and models trained with importance weighted samples (IW-LVAE). Note that the architecture of the DLGM is identical to that of SeRe-VAE; to ensure a fair comparison, the DLGM was given larger feature maps in the encoders to compensate for the additional bijective layer inputs in the SeRe-VAE. Therefore, *the performance benefits are solely attributed to the inclusion of the latent codes in subsequent stochastic layers in the hierarchy*. Our model outperforms the LVAE models, despite using a smaller latent dimensionality (128 vs. 100) and being trained with a single importance sample. Moreover, our model exhibits superior performance compared to the autoregressive IAF; this discrepancy could stem from the 1-layer architecture or the fact that a standard normal prior was used. This result indicates that a prior of equivalent expressive capacity communicating with the bijective layer could yield additional improvement. Finally, in our experiments the

| Model | Details | $log\, p(x) \geq$ |
|---|---|---|
| **Self-Reflective** | 10 layers / 10 variables each, diagonal Gaussian prior | **−81.17** |
| Importance Weighted Ladder | 5 layers / 128 variables total, #IW samples=10 | −81.74 |
| Ladder | 5 layers / 128 variables total | −81.84 |
| Self-Reflective IAF | 10 layers / 10 variables each, Standard Normal Prior | −81.96 |
| Inverse Autoregressive Flow | 1 layer / 100 variables, Standard Normal Prior | −83.04 |
| Deep Latent Gaussian Model | 10 layers / 10 variables each, diagonal Gaussian prior | −84.53 |
| Relaxed Bernoulli VAEs | 30 latent variables, exact factorization | −90 |

Table 1: Dynamically binarized MNIST Performance for VAEs without ResNet layers. 1000 importance samples were used for the estimation of the marginal likelihood. For the Ladder VAE performance, we refer to Table1 in Sønderby et al. (2016). The models were trained with a single importance sample unless otherwise noted (IW=1).

10-layer IAF took nearly *twice as long* to train compared to the SeRe-VAE. Finally, the Relaxed Bernoulli VAE (Webb et al., 2018) respects the factorization of the true posterior but scales up to 30 latent variables while not supporting recurrent refinement across layers. The learning curves, the architectural details and the training hyperparameters are provided in the appendix.

### 4.1.2 PERFORMANCE OF THE RESNET SERE-VAE

To demonstrate the capacity of our model when combined with complex layers, we replaced the MLPs with ResNets as in Salimans et al. (2015) while preserving the same number of latent variables. As shown in Table 2, our model performs better than all recent models that do not use expensive coupling or pixel-level autoregressive layers, either in the encoder or in the decoder, and on par with models of higher complexity. Especially for BIVA, it should be mentioned that more, 168 vs 100 of our model, latent variables are used. The full architectural details are provided in the appendix.

| Model | $log\, p(x) \geq$ |
|---|---|
| *Models with autoregressive (AR) or coupling (C) components* | |
| VLAE (Chen et al., 2017) | −79.03 |
| Pixel RNN (van den Oord et al., 2016b) | −79.20 |
| RQ-NSF (C) (Durkan et al., 2019) | −79.63 |
| Pixel VAE (Gulrajani et al., 2017) | −79.66 |
| RQ-NSF (AR) (Durkan et al., 2019) | −79.71 |
| IAF VAE (Kingma et al., 2016) | −79.88 |
| DRAW (Gregor et al., 2015) | −80.97 |
| Pixel CNN (van den Oord et al., 2016a) | −81.30 |
| *Models without autoregressive or coupling components* | |
| **SeRe-VAE** | **−79.50** |
| BIVA (Maaløe et al., 2019) | −80.47 |
| Discrete VAE (Rolfe, 2017) | −81.01 |

Table 2: Dynamically binarized MNIST performance for VAEs with sophisticated layers. 1000 importance samples were used for the estimation of the marginal likelihood. All performances listed here are taken from Maaløe et al. (2019) and Durkan et al. (2019). All models were trained with a single importance sample.

## 4.2 CIFAR10 NATURAL IMAGES

### 4.2.1 ABLATION STUDY

In this section, we study the effect of the different couplings between the layers of the architecture presented in Figure 1 on CIFAR-10 images which have dimension $(32, 32, 3)$. We consider a 16-layer architecture with each layer generating a $(8, 8, 3)$ patch of the image when partitioned in a spatial checkerboard pattern. We use $(8, 8, 2)$ latent spaces per layer. For the decoder, we use the mixture of discretized logistic distributions (Salimans et al., 2017). In particular, we investigate three different architectures:

- **case 1:** there are no couplings (no vertical edges) between the layers and each patch is independently generated from the others,
- **case 2:** there are couplings only between the decoders ($x_{l-1} \rightarrow x_l$ edges),
- **case 3:** there is feedback from the previous inference layer both in the observed space ($x_{l-1} \rightarrow x_l$ edges) and the latent space ($z_{l-1} \rightarrow \epsilon_l$, and $z_{l-1} \rightarrow z_l$ edges).

In all of the above cases, we consider joint bijective layers between the inference and generative network. One observation that we would like to make and turned out to be critical , when we tested our architecture on more complex regimes such as CIFAR-10, and in order to obtain significant predictive benefits from **case 3** compared to **case 2** was that we had to consider a lower bound for the variance in the prior layers. In other words, *a deep probabilistic should self-reflect by obtaining information from the previous inference layers but without being overly confident in its prior assumptions*. This can also be mathematically corroborated by examining the KL-divergence in the VAE objective of equation 2 for the residual parametrization introduced in section 3.4.2:

$$D_{KL}(q(\epsilon \mid z, x) \parallel p(\epsilon \mid z)) = \frac{(\delta\mu(x) - (1 - \delta\sigma(x))\mu(z))^2}{2\sigma^2(z)} + \frac{1}{2}\delta^2\sigma(x) - \frac{1}{2}\log(\delta^2\sigma(x)) - \frac{1}{2}. \tag{10}$$

As it can be seen, bounding $\sigma(z)$ from below prevents making the first term of the KL arbitrarily large. In these experiments, we take a unit lower bound.

| architecture | 128 epochs | 256 epochs | 512 epochs |
|---|---|---|---|
| **case 1** (no vertical edges) | 4.56 | 4.47 | 4.47 |
| **case 2** (coupled decoders) | 4.47 | 4.34 | 4.28 |
| **case 3** (SeRe-VAE) | 4.19 | 3.79 | 3.68 |

Table 3: Studying the impact in bits/dim of the connectivity between layers on the test set of CIFAR-10 data for a different number of training epochs. The KL was linearly annealed (S.II.A.1) from 0.2 to 1 for the first half of the training.

In Table 3, we observe that utilization of information from previous layers in the hierarchy both in the evidence space and in the latent space consistently improves inference. Moreover, the gap in the performance becomes larger as more training epochs are dedicated.

The attained performance could be further improved:

- *without increasing the complexity of the network* i) by re-distributing the latent variables allocated per-layer so that critical patches of the image are given more latent variables ii) further finetuning, especially of the lower bound of the scale in the prior iii) investigating block-coordinate descent optimization algorithms (with the parameters of each layer defining each block).
- *by increasing the complexity of the network*, in particular i) by deploying a deeper architecture ii) by increasing the receptive field of each inference layer so that it is coupled not only with the previous inference layers responsible for the generation of the immediately adjacent left/above patches iii) by employing recent deep VAE architectures for each one of the layer in our proposed scheme iv) by using more expressive, such as IAF, flows for the joint bijective layers v) by using pixel-autoregressive decoders.

Please note that none of the aforementioned suggestions introduces modeling redundancies (large latent spaces with many of their dimensions collapsing to their prior counterpart) or modeling mismatches between the true and the variational posterior.

### 4.2.2 Performance of a Self-Reflective, Variational Masked Autoregressive Flow on CIFAR-10

In this section, we introduce a *hierarchical latent variable normalizing flow*: the first VAE with a decoder consisting of normalizing flow transformations—realizing improvements over its purely generative counterpart. Due to space constraints we refer the reader to the appendix for a review of normalizing flows, as well as the full technical details of our architecture. A high-level description is provided here. The latent variables are generated by the proposed network shown in Figure 1. Subsequently, the latent variables $z$ are incorporated in the flow in two ways: i) conditioning the base distribution and ii) conditioning the bijective transformations. In the case of a Masked Autoregressive Flow (MAF) (Papamakarios et al., 2017) or an Inverse Autoregressive Flow (Kingma et al., 2016), the latter amounts to designing *conditional* MADE layers (Germain et al., 2015) that account for a mask offset so that the additional inputs $z$ are not masked out. The first amounts to building an amortized Gaussian layer. We used a 5 layer hierarchy of 40 latent variables each. We adopted a unit rank Gaussian base distribution in the decoder—parameterized as in Equation (9) in Rezende et al. (2014)—and diagonal Gaussian prior and posterior layers. We used neural spline bijective layers with coupling transformations (Durkan et al., 2019), which boosted the performance compared to affine transformations. We refer to our source code and the supplementary material for the implementation details. In Table 4, we compare against generative MAF models with the same or larger width, with or without training dataset augmentation with horizontal image flips and different number of MADEs. Our variational model exhibits significant improvement over the baselines.

| Model | Variational | #MADE layers | Width | Flipped Images | Test Loglikelihood |
|---|---|---|---|---|---|
| **SeRe-MAF** | **Yes** | **10 (2 flows, 5 layers)** | **1024** | **No** | $\geq$ **3190 (ELBO)** |
| MAF | No | 10 | 1024 | No | 2670 |
| MAF (5) (Papamakarios et al., 2017) | No | 5 | 2048 | Yes | 2936 |
| MAF (10) (Papamakarios et al., 2017) | No | 10 | 2048 | Yes | 3049 |

Table 4: Performance of different MAFs on CIFAR-10.

## 5 Conclusion and Discussion

In this paper, we presented self-reflective variational inference that suggests a structural modification for hierarchical VAEs (SeRe-VAE) and combines top-down inference with iterative feedback between the generative and inference network through shared bijective layers. This modification increases the representation capacity of existing VAEs, leading to smaller latent spaces and vast computational benefits without compromising the generative capacity of the model. We further introduced hierarchical latent variable normalizing flows which utilize the proposed architecture to recurrently refine the base distribution and the bijectors from the latent codes of the previous layer. For our experiments, we used uncoupled deterministic encoders; it would be interesting to explore any predictive benefits of a bottom-up deterministic pass of the inference network, especially for modeling natural images. The architecture could be further refined by adopting hierarchical stochastic layers. Finally, integration of pixel-regressive decoders and importance-weighted variations of the proposed scheme constitute directions for future research.

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
