# OpenReview forum: "Self-Reflective Variational Autoencoder"
_ICLR.cc/2021/Conference — Reject_

### Official Review · AnonReviewer4 · 2020-10-27
**Interesting and promising paper with potential for improvement in terms of clarity and experiments.**

**Rating:** 7
**Confidence:** 3

**Review:**

**Contributions & Significance**:
- The proposed paper introduces an encoder-decoder architecture for hierarchical VAEs based on bijective transformations, which preserves the true factorization of the posterior in its variational approximation (which is formally shown). (High)
- The resulting method shows improved performance on dynamically binarized MNIST compared to models of similar and higher complexity (autoregressive models). (Medium)
- A variation/extension of the model using normalizing flow transformations for the decoder shows significant improvement compared to Masked Autoregressive Flow (MAF) on CIFAR-10. (Medium)


**Pros**
- The proposed hierarchical architecture only relies on the latent code of the previous layer for conditioning, instead of all preceding latent codes such as in autoregressive models.
- The method shows improvement in terms of Log-Likelihood by rearranging the stochastic flow instead of relying on a computationally expensive architecture.
- The general method seems widely applicable for VAE-like models and as such can be combined with other architectural improvements or potential future work.

**Cons**
- The experiments are a bit limited in terms of the used datasets (Dynamically binarized MNIST & CIFAR-10).
- Sections 3.1.1-3.1.3 could be written more clearly. In particular I think the notation of the parameters is overly confusing (e.g. $\alpha_l$ and $\beta_l$ as functions). For 3.1.1 maybe just directly use $(\mu_l, \sigma_l)$?
- It is unclear how the number of data partitions L, their structure & order are selected. Does that have an effect, how large is this effect and what would the general recommendations be? What about the extreme cases (L=D, L=1)?

**Style**
In general the paper is well written and reasoned for.

**Minor Comments**
- The positioning of Figure 6 seems a bit awkward.
- Typo Figure 2: Prior Layers: I assume $p(\epsilon|z_2) $should be $p(\epsilon|z_1)$

**Summary**
The paper is well written, has a promising and sound approach and seems generally applicable. As such I vote for accepting it. However, it could still benefit from some improvement in terms of clarity and number of experiments (i.e. other data sets).

---

> ### Author Response · Authors · 2020-11-16
> **Thank your for your encouraging feedback and your constructive comments.**
>
> Please, find our response below:
>
> "The experiments are a bit limited in terms of the used datasets"
>
> We do agree that an ablation study on larger latent spaces could further illustrate the framework proposed in the paper (see also our response to Reviewer 1's comment D6.).  We also believe that there is vast potential for refinements of our model (such as hierarchical stochastic layers - in which case every single layer in our architecture can adopt advances on deep VAEs to generate the patch of the image for which it is responsible) which could tackle a larger input space. However, we feel it would become very complicated to describe all these refinements along with the core idea in a single work.
>
> "Sections 3.1.1-3.1.3 could be written more clearly. In particular I think the notation of the parameters is overly confusing"
>
> Please refer to our response to point D3.b of Reviewer 1. We involve the parameters $\alpha$, $\beta$ to convey the fact that the conditioning is achieved by rendering the parameters of the distribution a function of the conditioning factors. However, we will remove them when not necessary.
>
> "It is unclear how the number of data partitions L, their structure & order are selected. Does that have an effect, how large is this effect and what would the general recommendations be? What about the extreme cases (L=D, L=1)?"
>
> We have observed that increasing the number of layers increases the reconstruction capacity of the model. However, after some point the improvements are not significant enough (while increasing linearly -wrt the number of layers - the sampling time). We have chosen the number of layers in the experiments while keeping a trade-off  between training/sampling time and accuracy in mind. With only one layer, our model is equivalent to DLGM (no feedback to subsequent layers) and the performance matches the one presented in [1].
>
> "Typo Figure 2: Prior Layers:"
> Thank you for this correction. It should be $p(\epsilon_2|z_1)$.
>
> Please, do let us know if there are any other issues, that you would like to point out and you think could further highlight the potential of our work.
>
>
> [1] Rezende DJ, Mohamed S, Wierstra D. Stochastic backpropagation and approximate inference in deep generative models. arXiv preprint arXiv:1401.4082. 2014 Jan 16.

---

### Official Review · AnonReviewer3 · 2020-10-27
**Lack of clarity, novelty and empirical evidences**

**Rating:** 3
**Confidence:** 4

**Review:**

1. Summary
This paper proposes an augmentation of the Ladder VAE model (LVAE [1]) using 1. more flexible variational distributions using normalizing flows (denoted $f$) 2. an autoregressive component (because of the generative dependency $p( x_{l} | z_{l}, x_{l-1}) $ (*data layers*), although this is not directly stated as a contribution.
It is argued the proposed factorization of the inference network (top-down inference) matches the factorization of the true posterior, that follows the top-down design of the generative model. The authors recorded competitive performances on dynamically binarized MNIST.

2. [a] Strong Points
- the authors introduce a very flexible architecture for DGMs by combining recent improvements from the literature.

2. [b] Weak Points
- the paper overall lacks clarity: the idea could be explained in much simpler terms
- the contributions are weak: augmenting an LVAE model using flows brings little novelty
- the autoregressive structure introduced in the paper (which can be interpreted as an instance of state-space models [2]) is not well discussed and is contradictory with the claim that no *autoregressive layer* is used.
- an ablation study is required to disentangle the effect of 1. the flow augmentation and 2. the autoregressive structure
- aiming for a better inference network aims at tightening the variational bound, which should be measured empirically
- the diagonal Gaussian assumption for the variational family is not necessarily so limiting [3, 4] (as assumed in the paper).
- the results reported for CIFAR-10 are not competitive with sota models [3, 4] (SeRe-MAF) $\approx$ MAF (10) = 4.31 (original paper) $\gg$ 3.08 (BIVA [4]) > 2.91 (NVAE without flow [3]) .

3. Recommendation
Unfortunately, I recommend rejecting this paper.

4. Recommendation Arguments
SeRe-VAE combines multiple architecture improvements (top-down model (LVAE), flexible variational distributions using flows, autoregressive components) into a final model, which is tested using a single dataset (excluding cifar10 results which are not competitive) and without performing an ablation study. This prevents the community from understanding the effect of each of the architecture choices and does not guarantee that such an architecture could be effectively adapted to other contexts.

5. Questions to the Author
- Figure 1: why considering the prior independent? the generative model adopts a hierarchical structure $p(z_l  | z_{l+1})$

6. Feedback
Your work is an engineering prowess, I am saddened to recommend rejection. I think your work could greatly benefit from an ablation study and from defining the model in more minimal terms.

A few comments:
- The inference network is conditioned on $\mathbf{x}$, not the entire $\mathcal{D}$.
- you can measure the variational bound using the identity: $\operatorname{KL}(q(z | x) | p(z | x)) = log p(x) - \mathcal{L}(x)$
- please report CIFAR10 results in bit per dimension, as stated in the literature
- please report results on the more widely accepted Statistically binarized MNIST first, use dynamic MNIST as a second option.

[1] Sønderby, Casper Kaae, et al. "Ladder variational autoencoders." Advances in neural information processing systems. 2016.
[2] Fraccaro, Marco, et al. "Sequential neural models with stochastic layers." Advances in neural information processing systems. 2016.
[3] Vahdat, Arash, and Jan Kautz. "Nvae: A deep hierarchical variational autoencoder." arXiv preprint arXiv:2007.03898 (2020).
[4] Maaløe, Lars, et al. "Biva: A very deep hierarchy of latent variables for generative modeling." Advances in neural information processing systems. 2019.

---

> ### Author Response · Authors · 2020-11-16
> **Thank you for helping us clarify the novelty of our work and the significance of our contribution.**
>
> We address your concerns below:
>
> "the contributions are weak: augmenting an LVAE model using flows brings little novelty"
>
> The only common point between LVAE and Sere-VAE is the top-down inference (also adopted in NVAE, BIVA etc). The key points of our work are i) the shared bijective layers ii) feedback to both the prior and the posterior with latent factors of previous layers (motivated theoretically as a way to correct the factorization of the variational posterior). iii) partitioning of the evidence (image)  between the layers of the architecture, which casts  the problem of generating a large image to smaller subproblems while enabling feedback and recurrent refinement of the latent space. We also do *not* use flows. The term flow is usually used to describe a long chain of, potentially complicated, bijective transformations. We currently use a single, affine transformation.
>
>
> "the autoregressive structure introduced in the paper (which can be interpreted as an instance of state-space models [2]) is not well discussed and is contradictory with the claim that no autoregressive layer is used."
>
> Our architecture does not constitute an autoregressive structure. Usually, the term "autoregressive" is used to describe dependency on *all* previous pixels such that $p(x_i|x_0, x_1, ..., x_{i-1})$. We are instead conditioning on only a small portion of $x$ (patch $x_{l-1}$ ) generated by the previous layer and at once such that we model $p(x_{l}|x_{l-1})$ not $p(x_{l}|x_{1}, x_{2}, \dots, x_{l-1})$. More specifically, we do not model the distribution of $k$-th pixel of layer $l$ as a function of all previous pixels of layer $l$, or formally we do *not* consider $p(x^k_{l}|x_{l-1}, x^0_{l},  x^1_{l}, x^{k-1}_{l})$.
>
>
> "an ablation study is required to disentangle the effect of 1. the flow augmentation and 2. the autoregressive structure"
>
>  Thank you for this insightful suggestion. Without the coupling between the decoders (edges between x-s), the performance of the model on MNIST is 81.32 with MLP layers (please compare with Table 1) and 79.97 with ResNet layers (please compare with Table 2). We will include these experiments in the updated version of the manuscript. Without the common bijector, the correct factorization is no more feasible. The other cases in Tables 1 and 2 fall under this category.
>
>
> "aiming for a better inference network aims at tightening the variational bound, which should be measured empirically"
>
> The ELBO for the MLP case is 85.2 and for the ResNet case is 83.34 (you can also see the learning curves in S.II.A in the supplementary)
>
> "the diagonal Gaussian assumption for the variational family is not necessarily so limiting"
>
> Please note that in NVAE the latent space used is 5 (layers) * 4 x 4 (spatial dimensions) x 20 (channels) + 10 (layers) * 8 x 8 (spatial dimensions) x 20 (channels), which is orders of magnitude larger that the latent space we are considering (only 100 latent variables).  Our training time was less than a day on a single GPU. We compare with BIVA in Table 2 and we significantly outperform it with fewer latent variables.
>
> "the results reported for CIFAR-10 are not competitive with sota models "
> Please refer to our response to Reviewer 1. Moreover, variational MAF outperforms significantly MAF while offering a dimensionality reduction ( 200 - latent space-  vs 3072 - image space) alternative.
>
> "Recommendation Arguments SeRe-VAE"
>
> We agree that an ablation study on larger latent space will strengthen the impact of our work.
>
> "Figure 1: why considering the prior independent?"
>
> Figure 1 refers to DLGM (existing works), not our model. Please refer to Figure 2 for a high-level description of our work.
> We put these figures side-by-side to highlight differences between our proposed and relevant existing architecture.
>
> "The inference network is conditioned on $\boldsymbol{x}$, not the entire  $\boldsymbol{\mathcal{D}}$."
>
> We will update our notation, so that it refers to a single datapoint.
>
> "please report results on the more widely accepted Statistically binarized MNIST first, use dynamic MNIST as a second option."
>
> We do use  Statistically binarized MNIST, as all the papers in the literature.
>
> Please tell us if you have any other concerns on the clarity/novelty of our method which we can address in order to increase your confidence in the quality of this work.

---

> > ### Comment · AnonReviewer3 · 2020-11-20
> > **Thank you for your detailed reply, unfortunately my position remains the same**
> >
> > I am now more confident that the concepts presented in this paper have a scientific value. However, many issues remain, especially in terms of clarity (as also mentioned by Reviewer 1).
> >
> > "the contributions are weak: augmenting an LVAE model using flows brings little novelty"
> >
> > - I admit using the word "Flow" is rather provocative, yet a bijective affine transformation is a special case of Flows.
> > - The recurrent refinement (or autoregressive dependencies) is orthogonal to the claim of the "exact factorization"  (they could even be studied in two separate papers) and not clearly stated as a contribution, the idea is roughly introduced later in the paper.
> >
> > "the autoregressive structure introduced in the paper (which can be interpreted as an instance of state-space models [2]) is not well discussed and is contradictory with the claim that no autoregressive layer is used."
> >
> > Grouping the pixels into patches does not change the nature of the autoregressive factorization: $p(x_{l:l+m} | x_{<l})$. It is true that $p(x_l | x_{l-1})$ is a weaker factorization than $p(x_l | x_{<l})$, however this remains a special case of an autoregressive model. Therefore SeRe VAE uses autoregressive components and therefore the claim that "SeRe VAE does not use autoregressive layers" is incorrect. I think you mean that you do not use pixel-level level AR components (for which sampling is expensive), since the patch factorization requires L << D steps.
> >
> > "aiming for a better inference network aims at tightening the variational bound, which should be measured empirically"
> >
> > Measuring the quality of the variational bound involves approximating $\log p(x)$ and $\mathrm{KL}(q(z|x) | p(z|x))$ in my opinion.
> >
> > "Figure 1: why considering the prior independent?"
> >
> > I understand that figure 1 does not represent your work. Nonetheless, all hierarchical VAEs I have encountered feature a factorization $p(z_l | z_{l+1})`$.
> >
> > "please report results on the more widely accepted Statistically binarized MNIST first, use dynamic MNIST as a second option."
> >
> > The title of section 4.1 is "Dynamically binarized MNIST". This is confusing.

---

> > > ### Author Response · Authors · 2020-11-24
> > > **Thank you for your continued response!**
> > >
> > > "The recurrent refinement (or autoregressive dependencies) is orthogonal to the claim of the "exact factorization" (they could even be studied in two separate papers) and not clearly stated as a contribution, the idea is roughly introduced later in the paper"
> > >
> > > The distribution of the latent variables across the decoders (so that each layer is responsible for the generation of a different part of the data) is dictated by our motivation to respect the factorization. For example, if an edge z3->x1 was present in Figure 1, it would introduce a V-structure in the bayesian network, hence the path from z3 to z1 would no longer be d-separated. Due to this fact, and in order to keep the latent space small, the decoders utilize the evidence, not the latent variables (which would violate the correct factorization), that have already been generated by previous layers.
> > >
> > > "I understand that figure 1 does not represent your work. Nonetheless, all hierarchical VAEs I have encountered feature a factorization "p(zl|zl+1)"
> > >
> > > We have chosen to illustrate DLGM in Figure 1, which does not introduce conditional dependencies in either the prior or the posterior, since we feel is closest to our architecture (because of the generative layers that are shared between the prior and posterior layers). As you correctly pointed out, other works do indeed consider conditional priors and/or posteriors. However, these conditioning factors are not theoretically justified typically resulting in a deluge of latent variables. We cite and compare with these works.
> > >
> > >
> > > "Grouping the pixels into patches does not change the nature of the autoregressive factorization ..."
> > >
> > >  We agree that strictly speaking, the decoders are autoregressive. We corrected our wording in the updated version of the paper.
> > > However, this does not affect the performance of sampling. It requires L steps, which is the case for all deep-VAE architectures. Moreover, please note that the size of each decoder is reduced (since it generates 1/L-th of the image). Therefore, computational benefits are obtained not only from the L vs D (D== size of images) steps for the sampling but also from the reduced size of the conditioning factors. Finally, and given your broad definition of autoregression, please note that the deep VAE architectures that introduce conditional dependencies between the priors/and or posteriors (as you pointed out) are also autoregressive *in the latent space*  (while also claiming themselves non-autoregressive). Given the vast latent space (much larger than the data distribution of interest) they consider, this consideration induces significant computational overhead.
> > >
> > > "Measuring the quality of the variational bound involves approximating  in my opinion"
> > >
> > > We provided the ELBO in order to compare with the marginal likelihoods (estimated by importance sampling) provided in Tables 1, 2 to get a sense of the tightness. We also provide the KL in the appendix (Figure S.5)

---

### Official Review · AnonReviewer1 · 2020-10-27
**Possibly an interesting idea, but the paper is hard to follow**

**Rating:** 5
**Confidence:** 5

**Review:**

**GENERAL**
The paper proposes a new hierarchical architecture for VAEs. The main idea is to use invertible components that could be shared by the encoder and the decoder. However, the paper is very confusing in many parts, and the experimental studies are not too strong.

**Strengths:**
S1: The authors propose a new architecture for stochastic layers in VAEs.

S2: The idea is to use bijective layers shared by the generative and the variational parts of the VAE.

**Deficiencies:**
D1: Please correct Figures 1-5. First, the text seems to be "squashed" that hinders readability. Moreover, some words blend with each other. In Figure 5, rotating the text makes it almost impossible to read. Additionally, I believe including colorful squares or rectangles does not help; on the contrary, it makes it harder to read.

D2: The hierarchical model is presented through Figures (e.g., Figure 1) that is very hard to follow. I believe the authors have some interesting ideas, but it is completely overshadowed by rather unreadable and pretty unclear figures. It would be much better to simply express the model mathematically. At the moment, I do not follow what is the semantics of nodes, edges and colors.

D3: The text is hard to follow as well. For instance, the authors write that some components are implemented by ResNet or MLP. However, when MLP is used, and when ResNet is used? These statements are very confusing. Moreover, the authors introduce multiple notation for the same quantities that differ by one symbol, e.g., p(\epsilon_l | ...) and p(\epsilon ; ...). What is the purpose of using "|" and ";" interchangeably?

D4: The section about Residual Data Layers is written in a confusing manner. The authors introduce a two-step procedure to calculate \gamma. First, they compute an estimation of it. Then, they calculate a second estimation \delta \gamma_l. Then, they combine these two estimations to define \c_l^{\gamma}. Why do the authors mention estimations? Why are the two quantities summed? It is totally unclear and written in a clumsy fashion.

D5: The statements in Section 3.4 are very hand-wavy. The authors claim, e.g., that "The model, albeit hierarchical, is less prone to posterior collapse, since each layer is responsible for the generation of a different portion of the data". How do we know that? It is not obvious that this is the case.

D6: I appreciate the experiments provided by the authors. However, the provided comparison on CIFAR10 is hardly acceptable. First, currently using Normal distribution for the decoder is not widely used. Second, MAF is a great idea and an extremely interesting paper, however, nowadays it cannot be treated as a strong baseline.

*AFTER REBUTTAL* I would like to thank the authors for their rebuttal. I increase my score to 5. I am still unsatisfied with the presentation of the idea, because it is still rather hard to follow.

---

> ### Author Response · Authors · 2020-11-16
> **Thank you for your review!**
>
> D1, D2: We will take care of these minor issues in the updated version of the manuscript. In the Figures, the nodes represent random variables, the edges represent conditional dependencies. We have chosen colorful squares to match visually the random variables with the layers in Figures 4, 5 which are responsible for their generation.
>
> D3: a. "For instance, the authors write that some components are implemented by ResNet or MLP. However, when MLP is used, and when ResNet is used?"
> In the experiments in Section 4.1.1, we use only MLP layers. In Sections 4.1.2, 4.1.3, we use ResNet layers. In the Supplementary (Table S2, Table S3, Table S5), we provide all the hyper parameters for the experiments.
> b.  We use the notation $p(\boldsymbol{\epsilon}|\boldsymbol{z};\boldsymbol{\alpha})$ to highlight the fact that $\boldsymbol{z}$ is a conditioning factor for $\boldsymbol{\epsilon}$ in a distribution parametrized by $\boldsymbol{\alpha}$. We use the notation $p(\boldsymbol{\epsilon}|\boldsymbol{\alpha}(\boldsymbol{z}))$ to explain the amortized distributional layers and to highlight the fact that the aforementioned conditioning is achieved by rendering the parameters a function of the conditioning random variables. Please, do let us know if this is still not clear. We could also keep only one of the two notations in an updated version of the mauscript. Please, let us know which one you think is more clear.
>
> D4. A residual Gaussian distribution conditioned on two latent factors $\boldsymbol{y}, \boldsymbol{z}$ has the following parametrization:
> $$ p(\boldsymbol{x}|\boldsymbol{y},\boldsymbol{z} )= \mathcal{N}(\boldsymbol{\mu}(\boldsymbol{y}) \delta\boldsymbol{\sigma}(\boldsymbol{z})+\delta\boldsymbol{\mu}(\boldsymbol{z}),\delta\boldsymbol{\sigma}(\boldsymbol{z})\boldsymbol{\sigma}(\boldsymbol{y})) $$
> which can be interpreted as: the first distribution $\mathcal{N}(\boldsymbol{\mu}(\boldsymbol{y}), \boldsymbol{\sigma}(\boldsymbol{y}))$ is corrected, in light of the additional conditioning factor $\boldsymbol{z}$ that provides the corrections $\delta\boldsymbol{\sigma}(\boldsymbol{z})$, $\delta\boldsymbol{\mu}(\boldsymbol{z})$, in an affine manner. In case $\boldsymbol{z}$ does not provide further information on $\boldsymbol{x}$ (formally, $p(\boldsymbol{x}|\boldsymbol{y},\boldsymbol{z} )=p(\boldsymbol{x}|\boldsymbol{y}))$ the two corrections would collapse to 1 and 0 respectively. Please note, that in case of a discretized logistic distribution, the same affine correction can be considered. In the simpler Bernoulli case, we adopt a logit-based parametrization, hence the additive correction. We refer to $p(\boldsymbol{x}|\boldsymbol{y})$ as an estimation, because the corrections $\delta\boldsymbol{\mu}(\boldsymbol{z})$, and  $\delta\boldsymbol{\sigma}(\boldsymbol{z})$ can only increase the likehood. You may refer to Figure S.4 in the supplementary, where we qualitatively illustrate the effect of the additional distributional layer (correcting the first one).
>
> D5. Intuitively, this happens inherently and by construction of the model: the latent factors of each layer generate a different patch $\boldsymbol{x}_i $ of the image. Therefore, the latent variables would collapse to the prior only if the patch of the image itself was noise. To support this claim experimentally, we also provide in the Supplementary material (Figure S.5) the KL divergence for the latent variables per layer.
>
> D6. Please, note that the scope of the paper is to introduce a new principle for variational inference that i) enforces correct factorization in a deep probabilistic model and in an efficient manner ii) enables feedback to subsequent variational factors through shared bijective layers and amortized prior layers and not to offer SOTA results. We have intentionally used a simple (Gaussian) decoder to demonstrate that the predictive improvements come exclusively from the latent factors that are inserted in the model in a principled manner. Moreover, we think that compressing  (32,32,3) (as opposed to current deep VAE architectures which blindly insert latent variables resulting in latent spaces *much* larger than the image itself) images by using only 200 latent variables while still achieving competitive likelihood is no small feat. Furthermore, please do note that our architecture aims at refining the base distribution from the latent codes, hence making it applicable to all normalizing flows (not necessarily MAF).  However, we do agree that i) an ablation study on larger latent spaces (> 200 latent variables ) would strengthen our work. ii) Our work has the potential to reach SOTA results in a computationally efficient manner. However, we feel this would require further changes in the model (such as adopting hierarchical stochastic layers) and the inclusion of IAF bijectors, and we leave this as future work.
>
> Please tell us if you have any other concerns which we can clarify in order to increase your confidence in the quality of this work.

---

> > ### Comment · AnonReviewer1 · 2020-11-23
> > **After the rebuttal**
> >
> > Dear authors,
> >
> > Thank you for your detailed rebuttal. I highly appreciate the hard work you put into the rebuttal.
> > I still have some doubts, and I think it is due to rather a bit unclear presentation of the idea. However, I am (partially) satisfied with the rebuttal, and, therefore, I increase my score to 5.
> >
> > Best.

---

> > > ### Author Response · Authors · 2020-11-24
> > > **revised manuscript that incorporates all your suggestions on presentation has been uploaded**
> > >
> > > Thank you for your reply and for increasing your score! We have now uploaded an updated version of the manuscript that incorporates your feedback regarding the presentation. We feel that these changes significantly improve the clarity of the main ideas of the paper. In particular,
> > >
> > > 1) We have removed the dual notation of the amortized layers. The same concept is now described in simpler terms in section 3.4.1 of the revised manuscript.
> > >
> > > 2) We removed all but one figure from the main text. We moved them in the supplementary and they are now accompanied by explanatory text.
> > >
> > > 3) We explained mathematically the incremental refinement of the conditional distributions by the residual terms in section 3.4.2 of the revised manuscript.
> > >
> > > 4) We provided mathematical justification for some claims in the paper (see proposition 1).
> > >
> > > 5) We also included an ablation study on CIFAR-10 to study the behavior of our model with larger latent spaces (and using a mixture of discretized logistics in the decoder).

---

### Author Response · Authors · 2020-11-24
**Summary of Response to Reviews**

We thank all reviewers for their thoughtful comments and suggestions.

We uploaded a significantly revised version of the manuscript which addresses the concerns shared across the reviewers. In particular,

A) In order to improve the clarity and presentation of the main ideas introduced in the paper:

1. We cleaned-up the notation of the amortized layers (section 3.4.1 in the revised paper)

2. We explained the concept of the residual distributions with more mathematical rigor (section 3.4.2 in the revised paper), and we removed the relevant figure completely. Moreover, we included Proposition 1, which justifies theoretically our claim: "Finally, the use of these layers can be viewed as a hierarchical application of thereparameterization trick(Kingma & Welling, 2014)which is now conducive to a closed-form computation of the KL-divergence" in the initial version of the manuscript.

3. We moved most of the figures to the supplementary material, in order to express the model with more detail in the main text .

B) We added an ablation study (section 4.2.1 in the revised manuscript) of architectures, with increasing connectivity between layers in the hierarchy that use a mixture of discretized logistic distributions in the decoder, on CIFAR-10 to study the behavior of our model on larger latent spaces.

---

> ### Comment · AnonReviewer4 · 2020-11-24
> **Improved Clarity**
>
> Thank you for the updated version.
>
> In my opinion it improved the clarity quite a lot - especially for the residual distributional layers, which were in the initial version more difficult to understand, both in its construction and utility.

---

### Comment · ~Anthony_L._Caterini1 · 2021-01-21
**Relationship to Continuously-Indexed Normalizing Flows?**

I just wanted to make the authors aware that a very similar model to this has already appeared at ICML 2020, and has indeed been publicly available since just after the previous ICLR conference submission deadline: "Relaxing Bijectivity Constraints with Continuously-Indexed Normalising Flows (CIFs)" (https://arxiv.org/abs/1909.13833).

I think there should at least be a discussion about the relationship between the two methods in the paper, most notably the fact that CIFs use a similar bijective structure with inference to build the model, which also ensures that the inference model has the correct factorization stucture.

---

> ### Author Response · Authors · 2021-01-21
> **please provide more details on your work**
>
> Thank you a lot for your reference. Could you please provide a more precise overview of your work?
> More precisely,
>
> 1) is your architecture hierarchical?
>
> 2) what is the dimensionality of the latent variables you are considering?
>
> 3) what is the computation time of your method (training time, sampling time, inference time) ?
>
> 4) what kind of bijectors (number of transformations needed?) are you using?
>
> In our work, the correct factorization was mostly motivated by the hierarchical structure that layer-wise partitions the input space. The computational benefits stem from the fact that each layer is responsible for generating a different part of the data (and the design of the distributions in such a way so that it is guaranteed that each conditioning factor improves the performance). The joint bijector in the context of VAEs was motivated by the fact that it does not introduce KL-penalty - Proposition I - while still offering better latent codes (without ruining the correct factorization of the hierarchical structure -- Lemma 1 which is straightforward to derive). In this work, we do not use expensive normalizing flows (just a single affine transformation of the Gaussians)

---

> > ### Comment · ~Anthony_L._Caterini1 · 2021-01-22
> > **Further details on the relationship**
> >
> > Thank you for your response. In terms of the relationship between CIFs and your work, we refer you to Figure 2 in the CIF paper. Essentially, a single-layer SeRe-VAE corresponds to CIF with the arrow from Z to U reversed. The hierarchical versions of these models then differ in that multi-layer CIFs are obtained by taking the distribution of Z to be itself a CIF, whereas multi-layer SeRe-VAEs are obtained by taking the distribution of U to be itself a SeRe-VAE. For both models, the bijective structure allows obtaining the correct factorization structure of the true posterior - for CIFs, see Proposition B.15 in [1].
> >
> > There are indeed other lower-level differences (including implementation approaches), but we believe this summarizes the relationship between these models, and that a comparison of SeRe-VAEs with CIFs is thus warranted. This would lead to various relevant questions - e.g. a SeRe-VAE is able to change the dimensionality of the output of the bijections used at each layer, whereas for CIFs this is fixed. On the other hand, CIFs benefit from the composition of bijections across layers, whereas in SeRe-VAEs the output of each bijection is fed into the next only indirectly via epsilon (U in CIF notation). It would be interesting to understand the implications of these differences.
> >
> > [1] https://arxiv.org/abs/1909.13833

---

### Decision · Program_Chairs · 2021-01-07
**Final Decision**

**Decision:**

Reject

**Comment:**

The paper proposes a variant of the hierarchical VAE architectures. All reviewers felt that the paper's clarity was lacking. While the authors made very significant improvements during the feedback phase, which were recognized by reviewers, the paper could use a revision that takes clarity into account from the ground up. I also think that the ablation studies should be expanded (if SOTA is not the goal, then science should be), e.g., compare to the setting in which q does not share the bijective layers with p.